# Aerosolized Harmful Algal Bloom Toxin Microcystin-LR Induces Type 1/Type 17 Inflammation of Murine Airways

**DOI:** 10.3390/toxins16110470

**Published:** 2024-11-01

**Authors:** Joshua D. Breidenbach, Benjamin W. French, Lauren M. Stanoszek, John-Paul Lavik, Krishna Rao Maddipati, Sanduni H. Premathilaka, David Baliu-Rodriguez, Bivek Timalsina, Vaishnavi Aradhyula, Shivani C. Patel, Apurva Lad, Irum Syed, Andrew L. Kleinhenz, Thomas M. Blomquist, Amira Gohara, Prabhatchandra Dube, Shungang Zhang, Dhilhani Faleel, Fatimah K. Khalaf, Dragan Isailovic, R. Mark Wooten, James C. Willey, Jeffrey R. Hammersley, Nikolai N. Modyanov, Deepak Malhotra, Lance D. Dworkin, David J. Kennedy, Steven T. Haller

**Affiliations:** 1Department of Medicine, College of Medicine and Life Sciences, University of Toledo, 2801 W. Bancroft, Toledo, OH 43614, USA; joshuab@lanl.gov (J.D.B.); benjamin.french2@rockets.utoledo.edu (B.W.F.); bivek.timalsina@rockets.utoledo.edu (B.T.); vaishnavi.aradhyula@rockets.utoledo.edu (V.A.); shivani.patel8@rockets.utoledo.edu (S.C.P.); apurva.lad@utoledo.edu (A.L.); andrew.kleinhenz@utoledo.edu (A.L.K.); prabhatchandra.dube@utoledo.edu (P.D.); shungang.zhang@rockets.utoledo.edu (S.Z.); fathimadhilhani.mohammedfaleel@rockets.utoledo.edu (D.F.); james.willey2@utoledo.edu (J.C.W.); jeffrey.hammersley@utoledo.edu (J.R.H.); deepak.malhotra@utoledo.edu (D.M.); lance.dworkin@utoledo.edu (L.D.D.); 2Department of Medical Microbiology and Immunology, College of Medicine and Life Sciences, University of Toledo, Toledo, OH 43614, USA; irum.syed@rockets.utoledo.edu (I.S.); r.mark.wooten@utoledo.edu (R.M.W.); 3Biochemistry and Biotechnology Group, Bioscience Division, Los Alamos National Laboratory, Los Alamos, NM 87545, USA; 4Department of Pathology, College of Medicine and Life Sciences, University of Toledo, Toledo, OH 43614, USA; lauren.stanoszek@utoledo.edu (L.M.S.); thomas.blomquist@utoledo.edu (T.M.B.); amira.gohara@utoledo.edu (A.G.); 5Department of Pathology and Laboratory Medicine, Indiana University School of Medicine, Indianapolis, IN 46202, USA; 6Department of Pathology, Lipidomics Core Facility, Wayne State University, Detroit, MI 48202, USA; maddipati@wayne.edu; 7Department of Chemistry and Biochemistry, College of Natural Sciences and Mathematics, University of Toledo, Toledo, OH 43606, USA; meegahamadiththegedara.premathilaka@rockets.utoledo.edu (S.H.P.); baliurodrigu1@llnl.gov (D.B.-R.); dragan.isailovic@utoledo.edu (D.I.); 8Department of Medicine, College of Medicine, University of Alkafeel, Najaf 54001, Iraq; kareem.khalaf@rockets.utoledo.edu; 9Department of Physiology and Pharmacology, College of Medicine and Life Sciences, University of Toledo, Toledo, OH 43614, USA; nikolai.modyanov@utoledo.edu

**Keywords:** microcystin, MC-LR, aerosol, harmful algal bloom, inflammation

## Abstract

Harmful algal blooms are increasing globally and pose serious health concerns releasing cyanotoxins. Microcystin-LR (MC-LR), one of the most frequently produced cyanotoxins, has recently been detected in aerosols generated by the normal motions of affected bodies of water. MC-LR aerosol exposure has been linked to a pro-inflammatory influence on the airways of mice; however, little is understood about the underlying mechanism or the potential consequences. This study aimed to investigate the pro-inflammatory effects of aerosolized MC-LR on murine airways. C57BL/6 and BALB/c mice were exposed to MC-LR aerosols, as these strains are predisposed to type 1/type 17 and type 2 immune responses, respectively. Exposure to MC-LR induced granulocytic inflammation in C57BL/6 but not BALB/c mice, as observed by increased expression of cytokines MIP-1α, CXCL1, CCL2, and GM-CSF compared with their respective vehicle controls. Furthermore, the upregulation of interleukins IL-17A and IL-12 is consistent with Th1- and Th17-driven type 1/type 17 inflammation. Histological analysis confirmed inflammation in the C57BL/6 lungs, with elevated neutrophils and macrophages in the bronchoalveolar lavage fluid and increased pro-inflammatory and pro-resolving oxidized lipids. In contrast, BALB/c mice showed no significant airway inflammation. These results highlight the ability of aerosolized MC-LR to trigger harmful airway inflammation, requiring further research, particularly into populations with predispositions to type 1/type 17 inflammation.

## 1. Introduction

Harmful algal blooms (HABs) are on the rise worldwide in regions experiencing water warming and eutrophication or an increase in nutrients due to runoff and wastewater mismanagement [1,2,3]. These are often overgrowths of cyanobacteria, also referred to as blue-green algae, which is an ancient phylum of photosynthetic bacteria. While cyanobacteria have an important role in the ecosystem, many species are also capable of producing toxins, named cyanotoxins, such as microcystins, cylindrospermopsin, anatoxins, saxitoxins, and nodularins [4,5,6,7].

In this study, we focus on microcystin-LR (MC-LR), as this is one of the most abundant and toxic congeners of microcystin and is often used as a surrogate for all microcystins in governmental advisories [8]. Animal exposure models for MC-LR have primarily investigated the oral and systemic exposure routes, which leads to hepatotoxicity, as demonstrated by increased liver weight, slight-to-moderate liver necrosis, and hemorrhaging in the exposed animals [9,10]. Furthermore, there is growing evidence that MC-LR is toxic to the kidneys [11]. Our own investigations have found that oral MC-LR exposure exacerbates hepatic injury in a murine model of non-alcoholic fatty liver disease (NAFLD) and that pre-existing colitis (DSS-induced) is worsened by MC-LR oral exposure [12,13,14,15].

Importantly, it was recently reported that HAB toxins can be aerosolized from lake water by bubble-bursting [16]. Survey reports and case studies describe detectable concentrations of microcystins in airway mucosa and irritation in subjects who have been near affected bodies of water for short periods; however, we are only aware of one study that has modeled MC-LR aerosol inhalation exposure [17,18,19,20,21]. Nose-only inhalation exposure to 260–265 μg/m^3^ for 0.5, 1, or 2 h each day for 7 days resulted in minimal-to-moderate multifocal degeneration, neutrophilic inflammation, and necrosis within the nasal cavity. Of note, the investigators were unable to confirm the delivery of the aerosol to the central or peripheral airways [20]. A subsequent study was reported in which 6–7-week-old male Swiss mice were exposed via intranasal instillation of 10 μL of 6.7 ng/kg MC-LR or distilled water (vehicle) control once a day for 30 days, after which significant increases in granulocytes were observed upon histological analysis of the lung tissue [22]. Similarly, in other studies of systemic exposure, the effects on the lungs were primarily granulocytic inflammation [23,24]. The consistent observation of granulocytic inflammation in pulmonary tissues following MC-LR exposure regardless of route of exposure suggests that an aerosol exposure may induce a similar response. Indeed, we recently reported that MC-LR aerosol exposures in a primary human airway epithelium model resulted in pro-inflammatory gene expression and the production of chemokines that attract neutrophils [25].

It remains uncertain whether MC-LR aerosol exposures in a whole organism model will have a pro-inflammatory impact on the airways and if said inflammation could be biased toward a type 1 or type 2 immune response. Understanding the type of immune response involved is crucial for estimating the risk associated with MC-LR aerosol exposure. Type 2-driven human asthma is typically medically controllable with corticosteroid treatment, whereas type 1-driven asthma does not respond to corticosteroid treatment and is often associated with worse outcomes [26]. Importantly, strain differences in mice can lead to a predisposition in the immune response. For example, it is well-established that C57BL/6 mice respond in a type 1 biased fashion, to the same provocation that would cause BALB/c mice to respond with type 2 [27,28,29]. In this current study, we took advantage of these strain differences to thoroughly investigate the potential pro-inflammatory impact of MC-LR aerosol on murine airways.

## 2. Results

### 2.1. Murine Strains Exposed to MC-LR via Aerosol Inhalation

To thoroughly characterize the immune responses to MC-LR and any biasing therein, we performed a strain and sex comparison with the type 1 immunity-prone C57BL/6 and the type 2 immunity-prone BALB/c mice. Male and female mice of each strain were exposed to MC-LR aerosol or vehicle by nose-only inhalation one hour a day for 14 days (Figure 1A). The estimated deposition of MC-LR to the lungs each day was 13 µg/m^2^ (see Section 2.1). Lung tissues were prepared for histology and molecular evaluation within one hour after the final exposure. Body weights were monitored over the course of the study. While slight decreases were found for all groups throughout the study, female C57BL/6 mice were found to lose significantly less weight than their respective vehicle control (Figure 1B).

### 2.2. Microcystin-LR Aerosol Inhalation Induces Upregulation of Type 1/Type 17 Inflammation-Related Proteins in Murine Lung

Cytokines and chemokines were measured in lung lysates of MC-LR and vehicle-exposed mice to investigate the type of inflammation involved. Grouping these analytes into type 1 (blue), type 2 (yellow), or mixed immunity (green), a significant increase in type 1 and mixed immunity was found in the MC-LR-exposed lung lysate of the C57BL/6 mice compared with the vehicle-exposed (Figure 2A). Specifically, in the male mice the upregulated markers for type 1 were IL-17A (FC = 6.0, *p* = 0.001), IL-12 (FC = 2.3, *p* = 0.003), and IFNγ (FC = 2.0, *p* = 0.007). The upregulated markers for mixed immunity were MIP-1α (FC = 8.6, *p* < 0.001), CXCL1 (FC = 5.4, *p* < 0.001), IL-1α (FC = 3.7, *p* < 0.001), MIP-2 (FC = 3.3, *p* = 0.002), CCL2 (FC = 3.3, *p* = 0.017), CCL17 (FC = 3.1, *p* < 0.001), GM-CSF (FC = 2.8, *p* < 0.001), and CCL22 (FC = 2.6, *p* < 0.001). These alterations were greater in the male than the female C57BL/6 mice. However, lung lysates of the females still contained upregulated type 1 [(IL-17A; FC = 5.2, *p* = 0.034), and (IFNγ; FC = 2.6, *p* = 0.018)] and mixed immunity markers [(MIP-1 α; FC = 5.3, *p* = 0.001), (CXCL1; FC = 4.7, *p* = 0.002), and (MIP-2; FC = 2.8, *p* = 0.014)]. While type 2 markers IL-4 (FC = 1.4, *p* = 0.040), and IL-5 (FC = 1.4, *p* = 0.018) were statistically significantly upregulated in the male C57BL/6 mouse lung, these increases were minimal and did not occur in the female mice. Importantly, exposures did not significantly increase cytokine or chemokine expression in the BALB/c mice, which are prone to type 2 immune responses (Figure 2B). In fact, slight but significant decreases were observed in MC-LR-exposed BALB/c mice compared with vehicle mice. These male BALB/c had decreased IL-17A (FC = 0.64, *p* = 0.036), and IFNγ (FC = 0.67, *p* = 0.014), while the female mice had decreased IL-13 (FC = 0.75, *p* = 0.029).

### 2.3. Microcystin-LR Aerosol Exposure Induces Airway Inflammation

There is no existing information concerning the gravitational effects on the deposition of MC-LR aerosol. Therefore, the left lung was formalin-fixed and serially sectioned (from dorsal to ventral) at a thickness of five microns in the coronal plane and histology from the ventral, medial, and dorsal regions was reviewed by a board-certified pathologist. Because there were no apparent differences between these sections, the medial region was taken as representative for histopathologic analysis. There was significantly increased inflammation found in the MC-LR-exposed lung compared with the vehicle, and this was more evident in the male compared with the female C57BL/6 mice (Figure 3A,B). Consistent with the molecular findings, the BALB/c lung sections did not contain noticeable inflammation. Immunohistochemical staining revealed increased macrophage infiltration in the MC-LR-exposed C57BL/6 lung, reaching significance in the male and trending in the female mice (Figure 3C,D). Red arrows indicate positively stained cells and terminal bronchioles show non-specific stain which was consistent across all samples (Figure 3C).

To further characterize the immune infiltrates, bronchoalveolar lavage (BAL) was performed. A separate set of male C57BL/6 mice was used to avoid complicating the histological examination. A significant increase in total cells was found in the lavage fluid recovered from the MC-LR-exposed mice compared with the vehicle-exposed mice (Figure 4A). Consistent with the upregulated chemokines (Figure 2), this was due to a significantly greater presence of neutrophils and macrophages (Figure 4B,E). While some eosinophils and lymphocytes were identified in select MC-LR-exposed specimens, these findings were not found to be significant (Figure 4C,D).

### 2.4. Microcystin-LR Aerosol Inhalation Alters the Lipid-Omic Profile of the Lungs

Lastly, the lipid profile of exposed lungs was evaluated due to its important and contemporary role in airway inflammation (Figure 5). Lipid-omics was performed by LC–MS revealing seven oxidized lipids which were significantly increased in MC-LR-exposed male C57BL/6 mouse lung compared with vehicle [(5(S),15(S)-DiHEPE; log2FC = 3.3, *p* < 0.001), (LXB4; log2FC = 3.1, *p* = 0.004), (20-COOH LTB4; log2FC = 2.6, *p* = 0.008), (11,12-DiHETrE; log2FC = 1.9, *p* = 0.011), (14,15-DiHETrE; log2FC = 1.5, *p* = 0.022), (19,20-DiHDoPE; log2FC = 1.4, *p* = 0.034), and (TXB2; log2FC = 0.93, *p* = 0.036)]. Three were found to be decreased [(14-HDoHE; log2FC = −1.5, *p* = 0.040), (13,14dh-15k-PGF2α; log2FC = −1.7, *p* = 0.040), and (15-keto PGE2; log2FC = −2.0, *p* = 0.053)]. These differential lipid abundances represent pro-inflammatory and pro-resolving mediators. All analyte measurements were normalized to total protein content and are reported as Log2 fold change between MC-LR-exposed lung tissues and vehicle controls. The results for all analytes are available in the Appendix A.

## 3. Discussion

Overall, these results support previous claims that MC-LR exposure to airways may be pro-inflammatory [22,23,24,30,31,32]. Here, for the first time, we report that inhalation exposure to MC-LR aerosol elicits an inflammatory response in mouse lung tissue. We also report the first sex and strain comparison in murine MC-LR exposures.

After MC-LR aerosol exposure daily for 14 days, lungs were collected within one hour after the final exposure and assessed. A molecular immune profile was found by the measurement of immune-relevant cytokines and chemokines. In this profile in C57BL/6 mice, we found increases in markers associated with type 1/type 17 inflammation and mixed immunity, while there were no or only slight increases in type 2 markers. To further dissect the type 1/type 17 and type 2 responses, BALB/c mice were also assessed after following the same regimen of MC-LR exposure. While C57BL/6 mice are prone to type 1 responses, BALB/c are prone to type 2 [27,28,29]. Therefore, a response in BALB/c mice would support a non-biased or mixed influence of MC-LR. However, the BALB/c mice did not respond with type 2 immunity, supporting a conclusion that MC-LR exposure drives a type 1/type 17 biased immune response.

Histological assessment supported the current literature in demonstrating that MC-LR is pro-inflammatory in murine airways after intranasal instillation [22]. Because we previously reported increases in macrophages in gastrointestinal tissues exposed to MC-LR and the activation of macrophages in response to MC-LR in vitro, we performed an IHC stain for F4/80^+^ macrophages in the lung tissue, demonstrating their infiltration [33].

Consistent with the evidence of immune infiltration in the lung tissue, bronchoalveolar lavage of a separate set of male C57BL/6 mice revealed increased immune cells in the airways. Importantly, these immune cells (neutrophils and macrophages) coincide with the cytokine and chemokine profile observed. Specifically, the chemokines found to be upregulated in the MC-LR-exposed C57BL/6 mice (MIP-1α, CXCL1, CXCL2, and CCL2) are well-documented as powerful chemoattractant molecules for neutrophils and macrophages [34]. Notably, CCL17 and CCL22 are often associated with type 2 immunity but are also reportedly produced by neutrophils or contribute to their recruitment indirectly [35,36]. Additionally, IL-1α is a pro-inflammatory cytokine capable of neutrophil recruitment [37,38]. IL-17 is the prototypical lymphokine produced by type 17 helper T cells and can even drive neutrophilic airway inflammation on its own when delivered intratracheally [39,40,41]. Finally, IL-12 leads to the differentiation of type 1 helper T cells, which go on to activate macrophages [42].

Sexual dimorphism of the immune system is an important area of investigation and may be evident in the findings of this work in which the overall inflammation was less in the female than the male mice [43]. Although the dose was adjusted for the size of the mice, differing sizes likely impact the diameter of their airways, and, therefore, the basic transport phenomena may account for these differences as the age-matched females were slightly smaller than the males. Regardless, because of these and the strain comparison results, the bronchoalveolar lavage and lipid-omics analyses were only performed in male C57BL/6 mice.

One driver of the increased unresolved inflammation from MC-LR can be explained through the understanding of the critical physiological and pathological mediators of inflammation: oxidized lipids. In fact, in a model of endotoxin-induced lung injury in mice, inhibition of arachidonic acid (AA) release from the phospholipid membrane via inhibition of phospholipase A2 (PLA2) prevented lung injury. This supports a model of rapid-onset increased capillary permeability mediated via the PLA2 pathways instead of, or in addition to, the common paradigm of cytokine-induced inflammation [44,45]. The increased capillary permeability can then lead to the release of cytokines that propagate further lung injury [44,46,47]. In this way, PLA2 primarily mediates inflammation via catalyzing the hydrolysis of phospholipids to AA. AA is further metabolized by cyclooxygenase (COX), lipoxygenase (LOX), and cytochrome P-450 (CYP450) into prostaglandins (PGs), thromboxanes (TXs), leukotrienes (LTs), and epoxyeicosatrienoic acids (EETs).

Inflammation can be mediated by an increase in the biosynthesis of inflammatory mediators such as LTB4 or PGE2 and/or exacerbated by an inhibition of their catabolism. The presence of downstream, less active metabolites of LTB4 (20-COOH LTB4) (Figure 5) suggests the inflammatory lipid biosynthesis triggered by neutrophils but also an attempt by the system to mitigate the inflammation [48,49]. Increased LXB4 and 20-COOH LTB4 demonstrate a yin and yang effect where the macrophage and neutrophil mediated-inflammation elicited by LTB4 is directly opposed by pro-resolutory, granulocyte-inhibitory LXB4 [50,51,52,53]. Epoxy fatty acids are known anti-inflammatory lipids, and these are inactivated by hydrolysis to their corresponding dihydroxy fatty acids (DiHETrEs and DiHOPE) by epoxide hydrolase [54,55,56]. An increase in the DiHETrEs and DiHOPE also contributes to the exacerbation of inflammation by the inflammatory mediators in a process called unalamation [57]. In the present study, increased 5(s),15(s)-DiHEPE in the lung of MC-LR-exposed mice (Figure 5) is consistent with the increased eosinophils also observed (Figure 4), as this metabolite is a potent chemotactic agent for eosinophils [58,59,60]. An increase in TXB2 in the MC-LR mice compared to the vehicle mice also signifies increased inflammation, as TXB2 is a potent activator of platelet aggregation and cytokine storm [61,62,63,64]. Additionally, increased concentration of pro-inflammatory 11,12- and 14,15-DiHETrEs (Figure 5) may signify increased inflammation in the MC-LR mice, as these metabolites have been studied to be significantly associated with worsening hepatic injury and fibrosis [65,66]. Although they have not been studied in lung tissue, it is plausible to assume that their elevated levels may be playing a role in the progression of inflammation in MC-LR mice. Conversely, decreased 15-keto PGE2 in the exposed mouse lung (Figure 5) can be correlated to negative regulation of neutrophil-mediated inflammation, as PGE2 has potent anti-inflammatory effects [67,68,69]. Thus, the lipid-omic profile highlights a multidimensional system where MC-LR exposure leads to an inflammatory lipid mediator response (LTB4 and PGE2 biosynthesis), exacerbation of the inflammatory response (decreased catabolism of PGE2 and increased metabolism of epoxy fatty acids), suppression of resolution response (decrease in the SPM precursor 14-HDoHE), and an attempt to resolve the inflammation (LXB4 and 5(S),15(S)-DiHEPE biosynthesis) with the inflammatory state ultimately prevailing as evidenced by the inflammatory cytokine release.

The human relevance of these results is clear, as we have previously reported findings that are consistent with this pro-inflammatory effect in healthy human primary airway epithelium [25]. In this previous work, MC-LR was found to upregulate gene expression related to inflammation, such as chemokines, and conditioned media was found to attract primary human neutrophils.

## 4. Conclusions

In conclusion, we report that MC-LR aerosol exposure induces inflammation in the murine lung. Furthermore, this appears to be driven by a type 1/type 17 immune response, resulting in the attraction of neutrophils and macrophages to the airways. Taken together, we believe that this work further supports the need to assess the risk of inhalation exposure to harmful algal bloom toxins, especially in the context of type 1/type 17 inflammation.

## 5. Materials and Methods

### 5.1. Animals and Aerosol Exposures

Male and female C57BL/6 (C57BL/6J) and BALB/c (BALB/cJ) mice (000664 and 000651; The Jackson Laboratory, Bar Harbor, ME, USA) were 13–14 weeks old at the beginning of this exposure study. All mice were caged socially in microisolator cages, given access to food and water ad libitum, and kept on a 12:12 hour dark–light cycle. Protocols for animal experimentation were approved by The University of Toledo Institutional Animal Care and Use Committee (IACUC protocol #108663, approval date 9 February 2016).

Mice were exposed to MC-LR (item no. 10007188; Cayman Chemical, Ann Arbor, MI, USA) aerosol or vehicle (water or saline) aerosol using an inExpose (SCIREQ; Montréal, QC, Canada) animal exposure system equipped with a nose-only adapter and a small particle (2.5–4 µm) VMD Aeroneb Lab ultrasonic nebulizer. Exposures were one hour each day for 14 days, and tissues were harvested within one hour after the final exposure. Lung deposition of 13 µg/m^2^ per day was chosen to mimic mass deposition of MC-LR in our previous in vitro study [25]. The concentration of solution required to achieve the mass deposited were related by the following:$$mass\;deposited=\frac{\frac{\left(Q_{neb}\right)\left(C_{sol}\right)\left(DC\right)}{Q_{bias}}\left(V_{min}\right)\left(t\right)\left(Df\right)}{A}$$
where,

Q_neb_ = output rate of nebulizer = 0.3 mL/min

C_sol_ = concentration of MC-LR in solution

DC = duty cycle = 50%

Q_bias_ = bias flow = 3.5 L/min

V_min_ = minute volume of mouse = 0.04 L/min [70]

t = exposure time = 60 min

Df = deposition fraction = 0.2

A = surface area of mouse lung = 0.02 m^2^

### 5.2. Histology Preparation and Stain

The left lung of each mouse was fixed by gentle perfusion with 10% buffered formalin through the left bronchus before complete submersion in 10 mL of fixative for 24 h on a rocker. Afterward, the formalin fixative was replaced with 70% histological grade ethanol for at least 24 h before paraffin embedding. Whole lungs were sectioned serially (from dorsal to ventral) at a thickness of five microns in the coronal/frontal plane and stained by hematoxylin and eosin (H&E).

Immunohistochemistry stains for F4/80^+^ macrophages were adapted from a previous publication [71]. Briefly, slides were deparaffinized via incubation in 100% xylene for 10 min. Rehydration was facilitated by incubation in 100%, 90%, and 80% ethanol followed by DI water for 5, 3, 3, and 6 min, respectively. Epitopes were recovered by 5 min treatment with proteinase K (S302080-2; DAKO, Santa Clara, CA, USA). Slides were then washed for 10 min in phosphate-buffered saline (PBS). Endogenous peroxidase activity was quenched with a 30 min incubation with freshly prepared 0.3% H_2_O_2_ in methanol. Slides were washed again for 10 min in PBS and then blocked with 2.5% goat serum (Normal Goat Serum Blocking Solution, 2.5%; S-1012-50; Vector Laboratories, Newark, CA, USA) for 45 min. The slides were then treated with primary antibody [250 µL PBS + 5 µL serum (2.5% goat serum) + 2.5 µL primary antibody (F4/80 clone: A3-1; Bio-Rad, Hercules, CA, USA)] and [250 µL PBS + 5 µL 2.5% goat serum] on minus primary controls, for 90 min. The antibody was vacuumed off to prevent accidental mixing. Slides were washed again for 10 min in PBS before treatment with secondary antibody [500 µL PBS + 10 µL serum (2.5% goat serum) + 2.5 µL secondary antibody (biotinylated anti-rat goat IgG, BA-9401; Vector Laboratories, CA, USA)] for 30 min. The slides were washed again for 10 min in PBS before incubation with horseradish peroxidase (HRP) kit [1 mL PBS + 20 µL Reagent A + 20 µL Reagent B] from the VECTASTAIN ABC kit (VECTASTAIN Elite ABC-HRP Kit; PK-6101; Vector Laboratories, CA, USA) for 30 min. Slides were washed again for 10 min in PBS. DAB substrate (DAB Substrate Kit, HRP; SK-4100; Vector Laboratories, CA, USA) was prepared without nickel and added while observing precipitate development under microscope (roughly 6–10 min). The reaction was stopped by washing in a large volume of DI water. Slides were counterstained for 2 s in Harris-modified hematoxylin followed by 10 dips in differentiation agent [acid alcohol; 2 mL glacial acetic acid in 400 mL ethanol] and finally 10 dips in bluing agent [ammonium water; 1 mL ammonium hydroxide in 500 mL DI water]. Finally, dehydration was performed by reverse incubations through ethanol gradient and finished in xylene. For best results, each PBS wash should begin with 5 dips and then a 10 min incubation in fresh PBS and all steps should be performed at room temperature with room temperature reagents.

Images of all stained slides were collected at 20X on a VS120 Virtual Slide Microscope (Olympus, Tokyo, Japan).

### 5.3. Histology Scoring

Histology slides of H&E-stained lung were graded by a board-certified, blinded pathologist following a published method [72]. Briefly, interstitial/perivascular inflammation, intra-alveolar inflammation, endothelialitis, and bronchitis were graded on a scale of 0–3, with 0: absent, 1: mild, 2: moderate, 3: severe. A total inflammation score was then found as a sum.

Immunohistochemical stains were scored via automated image analysis adapted from a method for evaluating fibrosis [73,74]. Using features built-in to Image-Pro Plus 7.0.1 (Media Cybernetics, Rockville, MD, USA), a macro was trained to process images of whole lung sections stained for F4/80^+^ macrophages, as described above, to enumerate the positively stained cells or the whole tissue area. The numbers of detected cells throughout each whole section were then divided by the calculated whole tissue area to achieve a normalized macrophage count for each specimen.

### 5.4. Bronchoalveolar Lavage

In a separate cohort of mice that were used for histology, bronchoalveolar lavage was performed and adapted from a published method [75]. After euthanasia by exsanguination under isoflurane anesthesia, airways were lavaged with sterile normal saline (0.9% NaCl). Briefly, the trachea was exposed by midline incision and cannulated with a custom device [a 23 G needle inserted into a segment of transparent plastic polyethylene 21 G tubing, (inner diameter: 0.58 mm, outer diameter: 0.965 mm, and length: 0.5 cm)]. A segment of thin polyester thread was tightened around the trachea to hold the cannulated device in place and assist the seal. Over the course of 30–45 s, 1 mL of sterile normal saline was injected. The bronchoalveolar lavage fluid (BALF) was then aspirated over the course of 30–45 s while gently pumping the thorax. The fluid recovered was centrifuged at 200× *g* for 10 min and red blood cells were lysed by 3 min incubation with ACK lysing buffer (A1049201; Thermo Fisher; Waltham, MA, USA). After washing with 10 mL PBS, cells were resuspended and total cell counts were obtained by TC20 automated cell counter (Bio-Rad; Hercules, CA, USA). For BALF differential cell counts, slides were prepared by cytospin (2000 RPM for 3 min) and stained 24 h later with Hema 3 Manual Staining System (22-122911; Fisher Scientific; Hampton, NH, USA). In total, 200 cells per slide were assessed by a board-certified, blinded pathologist who categorized each cell type based on various features including size, shape, and nuclear/cytoplasmic characteristics. All cell counts were normalized by the recovered BALF volume of each animal.

### 5.5. Protein Measurements

Chemokines and cytokines were evaluated at the protein level from lung lobe lysates of exposed and vehicle control mice. Whole lung lobes were placed in 2 mL round-bottom centrifuge tubes with 1 mL DMEM cell medium (DML28; Caisson Laboratories, Inc., Smithfield, UT, USA) along with a stainless steel bead. Tissues were mechanically lysed using a TissueLyser II (QIAGEN; Hilden, Germany) at 25 Hz for 10 min. Protein measurement services were performed by Quansys Biosciences (Logan, UT, USA). Custom multiplex ELISA-based Q-Plex™ technology delivered quantification of murine protein cytokines and chemokines: IL-1α, IL-1β, IL-2, IL-3, IL-4, IL-5, IL-6, IL-10, IL-12p70, IL-13, IL-17A, MCP-1 (CCL2), IFNγ, TNFα, MIP-1α (CCL3), GM-CSF, RANTES (CCL5), Eotaxin (CCL11), MIP-2 (CXCL2), KC (CXCL1), MDC (CCL22), TARC (CCL17), TCA-3 (CCL1). For results of all analytes, please see the Appendix A. All protein concentrations were calculated as pg analyte per mg total protein.

### 5.6. Lipid-Omics by Mass Spectroscopy

Lung lobe homogenate samples were spiked with internal standards (5 ng each of 15(S)-HETE-d8,14(15)-EpETrE-d8, Resolvin D2-d5, Leukotriene B4-d4, and Prostaglandin E1-d4) for recovery and quantitation. Polyunsaturated fatty acid metabolites were extracted using C18 columns and analyzed by LC–MS as described previously [76,77,78]. Briefly, HPLC was performed on a Prominence XR system (Shimadzu; Kyoto, Japan) using a Luna C18 (3 μ, 2.1 × 150 mm^2^) column (Phenomenex; Torrance, CA, USA). Via electrospray ionization, the HPLC eluate was delivered to a QTRAP5500 mass analyzer (ABSCIEX, MA, USA) run in negative ion mode with the following conditions: curtain gas: 35 psi, GS1: 35 psi, GS2: 65 psi, temperature: 600 °C, ion spray voltage: −1500 V, collision gas: low, declustering potential: −60 V, and entrance potential: −7 V. The data were collected using Analyst 1.6.2 and analyzed using MultiQuant software 3.0.3 [both from Sciex (Framingham, MA, USA)]. Each analyte was normalized against the total protein in each sample and was reported as ng/mg protein and is displayed as the log base 2 of the fold change (log_2_FC) between MC-LR-exposed and vehicle control.

### 5.7. Statistics

The statistical tests in this work were completed in GraphPad Prism version 7.0.5 for Windows (GraphPad Software 7.0.5; San Diego, CA, USA). All comparisons were made between two groups, the MC-LR-exposed and the respective vehicles. In all cases, unpaired Student’s *t*-tests were applied and significance is demarcated as a *p*-value < 0.05 and marked as *, **, ***, **** indicating *p* ≤ 0.05, 0.01, 0.001, 0.0001, respectively.

## Figures and Tables

**Figure 1 toxins-16-00470-f001:**
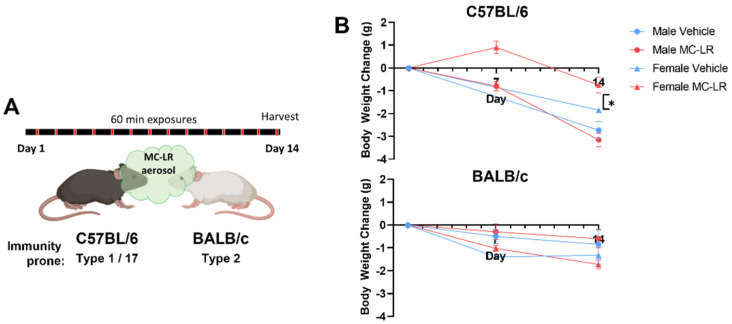
Overview of MC-LR aerosol exposures. (**A**) Timeline of the exposure study. (**B**) Body weight changes over the course of the 14-day study displayed as grams compared with the initial body weights of each animal. n = 4; Student’s *t*-test between MC-LR-exposed (red) and respective vehicle (blue), * indicates *p* ≤ 0.05.

**Figure 2 toxins-16-00470-f002:**
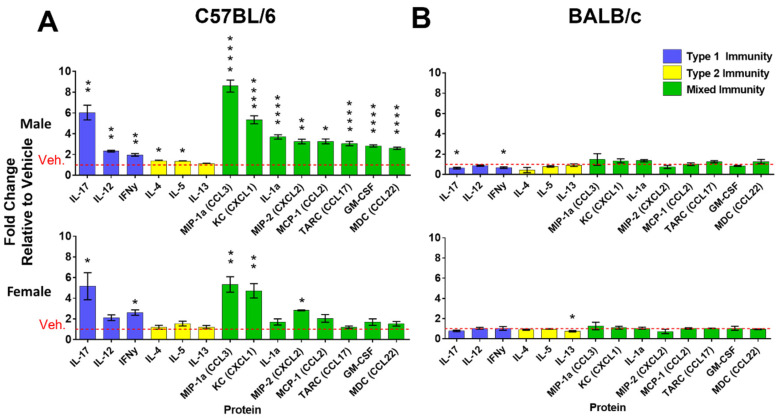
Molecular markers of inflammation in the lung tissue. Concentration of cytokine and chemokine proteins from lung lysate of male and female C57BL/6 (**A**) and BALB/c (**B**) mice. Values indicate fold change (FC) in concentrations from MC-LR-exposed mice compared with vehicle-exposed mice. Red dashed line indicates unchanged (FC of 1). Type of immunity associated with protein analyte: type 1 (blue), type 2 (yellow), mixed (green). *, **, **** indicates *p* ≤ 0.05, 0.01, 0.0001, respectively. Statistics by Student’s *t*-test between MC-LR-exposed and respective vehicle controls. n = 4 in all measurements.

**Figure 3 toxins-16-00470-f003:**
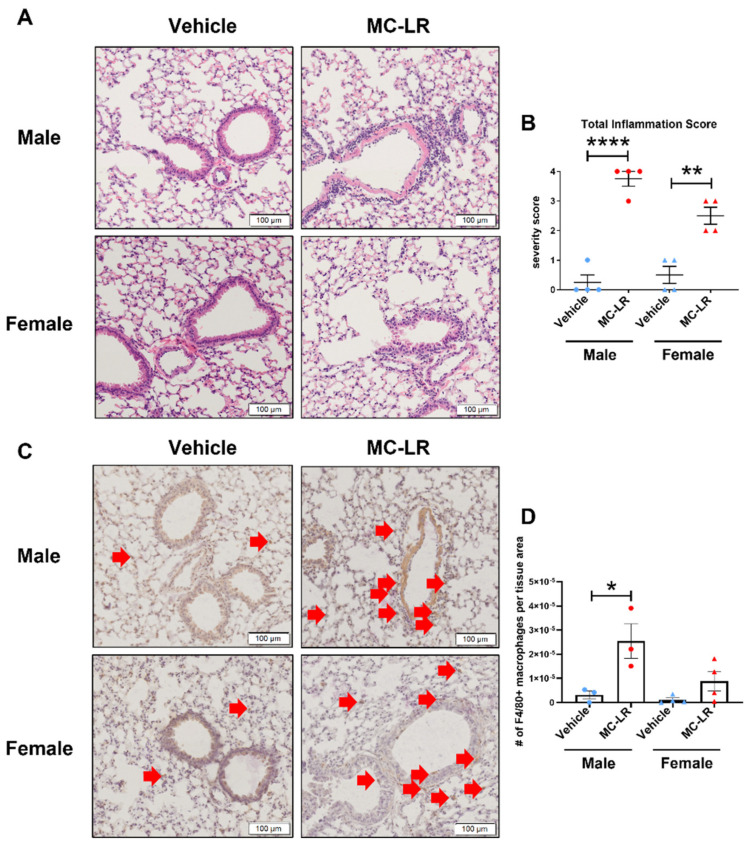
Histological evaluation of the lung. (**A**) Representative images of H&E-stained sections of male and female C57BL/6 mouse lung after MC-LR or vehicle aerosol exposure. (**B**) Pathologist’s scoring of inflammation severity in the H&E-stained sections. (**C**) Representative images of sections stained by IHC for F4/80 (a murine macrophage marker) in male and female C57BL/6 mouse lung after MC-LR or vehicle aerosol exposure. Red arrows indicate positively stained cells. (**D**) Results of an image analysis software-assisted enumeration of F4/80+ cells, normalized to the tissue area of each section. *, **, **** indicates *p* ≤ 0.05, 0.01, 0.0001, respectively. Statistics by Student’s *t*-test between MC-LR-exposed and respective vehicle controls.

**Figure 4 toxins-16-00470-f004:**
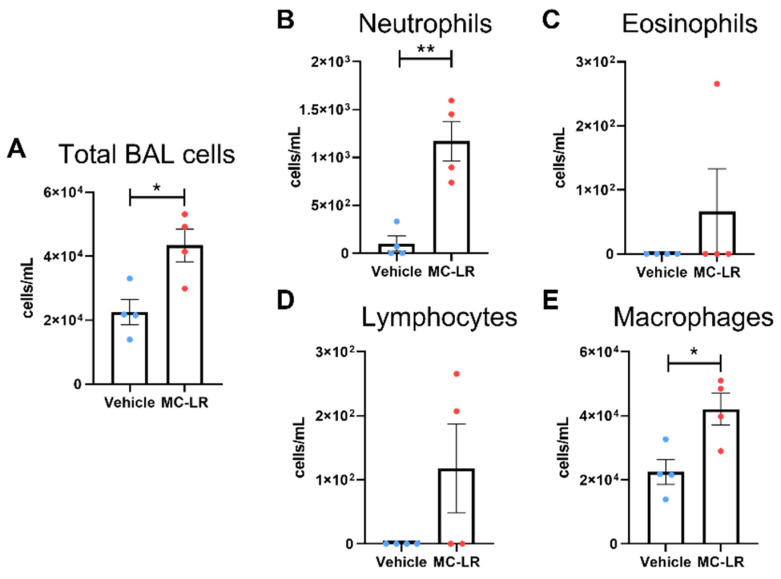
Bronchoalveolar lavage fluid analysis. Cellular constituents of BAL fluid normalized to the recovered volume. (**A**) total cells, (**B**) neutrophils, (**C**) eosinophils, (**D**) lymphocytes, and (**E**) macrophages. * and ** indicate *p* ≤ 0.05 and 0.01, respectively. Statistics by Student’s *t*-test between MC-LR-exposed and respective vehicle controls.

**Figure 5 toxins-16-00470-f005:**
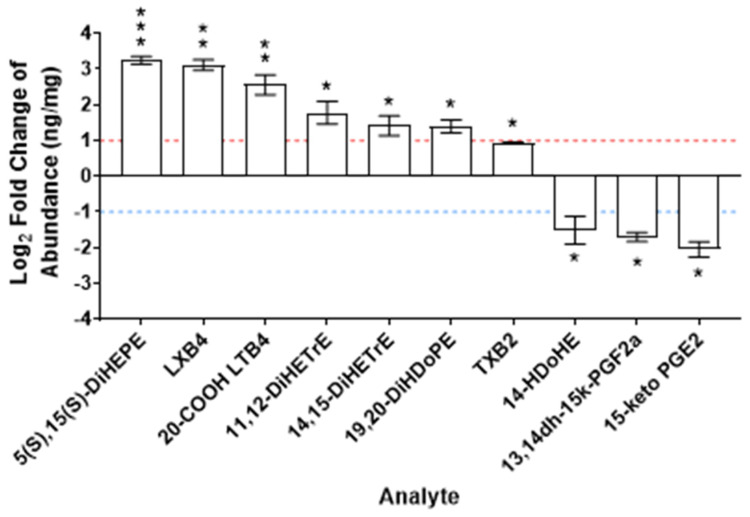
Lipid markers of inflammation in lung tissue. Concentration of arachidonic acid metabolites from lung lysate of male C57BL/6 mice. Values indicate log2FC in concentrations from MC-LR-exposed mice compared with vehicle. Dashed lines indicate Log2FC of 1 or −1, which correspond with an FC of 2 or −2, respectively. *, **, *** indicate *p* ≤ 0.05, 0.01, 0.001, respectively. Statistics by Student’s *t*-test between MC-LR-exposed and respective vehicle controls. n = 4 in all measurements.

## Data Availability

The raw data supporting the conclusions of this article will be made available by the authors, without undue reservation. The Appendix A are available through a public repository.

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
