# Peer review of "Aerosolized Harmful Algal Bloom Toxin Microcystin-LR Induces Type 1/Type 17 Inflammation of Murine Airways"

_toxins, 2024, doi:10.3390/toxins16110470_

Round 1
Reviewer 1 Report
Comments and Suggestions for Authors
The authors have reported that aerosolized cyanobacterial toxin (MC-LR) induces pro-inflammatory responses and immune cell infiltration in lung tissues of mice biased to type 1/type 17 inflammation in a strain dependent manner. The findings underscore the health risks of MC-LR, particularly for individuals predisposed to or with pre-existing type 1/type 17 inflammation. The manuscript is well-prepared but requires minor improvements.
Some general comments:
The introduction section is too brief and should include important information about the study.
The discussion of the LC-MS data in the results section is insufficient.
The statement "BALB/c mice showed no significant airway inflammation" needs to be supported by relevant literature in the results section.
Additionally, a table comparing findings from different studies should be added.
Comments on the Quality of English LanguageMinor changes are recommended.
Reviewer 2 Report
Comments and Suggestions for Authors
Comments to the Author
1. You observed a strong type 1/type 17 immune response in C57BL/6 mice but not in BALB/c mice. Can you elaborate on the potential mechanisms driving this strain-specific immune response to MC-LR aerosols?
2. Given that you used a 14-day exposure model, have you considered investigating the long-term or chronic exposure to MC-LR aerosols, and how this might affect airway inflammation over extended periods?
3. Your results indicate a more pronounced immune response in male C57BL/6 mice. Do you have any hypotheses on why male mice are more susceptible to MC-LR-induced inflammation than females?
4. Could the strain-specific responses observed in mice have implications for different immune predispositions in human populations exposed to aerosolized cyanotoxins? How might this research translate into potential human health risks?
5. Do you have any data or hypotheses on the molecular mechanisms by which MC-LR aerosol induces the observed type 1/type 17 inflammatory responses in the lungs?
6. Was there any attempt to explore the dose-response relationship in MC-LR aerosol exposure? How do you think varying concentrations of MC-LR would affect immune responses?
7. You mentioned increased pro-inflammatory oxidized lipids in the C57BL/6 mice. Could you explain how these lipids contribute to the inflammatory process and whether they play a critical role in the observed immune responses?
8. How do you expect these findings to compare to other mammalian species or even aquatic species that might also be exposed to aerosolized MC-LR? Could this model be useful for broader ecological studies?
9. Were there any unexpected histopathological changes in either mouse strain that you found particularly interesting? Did the inflammation affect any lung regions differently than expected?
10. Based on your findings, do you foresee any potential therapeutic strategies or preventive measures that could mitigate the pro-inflammatory effects of MC-LR aerosols in at-risk populations?
